# Sorghum Spike Detection Method Based on Gold Feature Pyramid Module and Improved YOLOv8s

**DOI:** 10.3390/s25010104

**Published:** 2024-12-27

**Authors:** Shujin Qiu, Jian Gao, Mengyao Han, Qingliang Cui, Xiangyang Yuan, Cuiqing Wu

**Affiliations:** 1College of Engineering, Shanxi Agricultural University, Jinzhong 030801, China; qiushujin@sxau.edu.cn (S.Q.); z20223672@stu.sxau.edu.cn (J.G.); 202430788@stu.sxau.edu.cn (M.H.); cqwu@sxau.edu.cn (C.W.); 2Dryland Farm Machinery Key Technology and Equipment Key Laboratory of Shanxi Province, Jinzhong 030801, China; 3College of Agriculture, Shanxi Agricultural University, Jinzhong 030801, China; yuanxiangyang200@sxau.edu.cn

**Keywords:** sorghum spike detection, YOLOv8s, Gold module, LSKA attention mechanism

## Abstract

In order to solve the problems of high planting density, similar color, and serious occlusion between spikes in sorghum fields, such as difficult identification and detection of sorghum spikes, low accuracy and high false detection, and missed detection rates, this study proposes an improved sorghum spike detection method based on YOLOv8s. The method involves augmenting the information fusion capability of the YOLOv8 model’s neck module by integrating the Gold feature pyramid module. Additionally, the SPPF module is refined with the LSKA attention mechanism to heighten focus on critical features. To tackle class imbalance in sorghum detection and expedite model convergence, a loss function incorporating Focal-EIOU is employed. Consequently, the YOLOv8s-Gold-LSKA model, based on the Gold module and LSKA attention mechanism, is developed. Experimental results demonstrate that this improved method significantly enhances sorghum spike detection accuracy in natural field settings. The improved model achieved a precision of 90.72%, recall of 76.81%, mean average precision (mAP) of 85.86%, and an F1-score of 81.19%. Comparing the improved model of this study with the three target detection models of YOLOv5s, SSD, and YOLOv8, respectively, the improved model of this study has better detection performance. This advancement provides technical support for the rapid and accurate recognition of multiple sorghum spike targets in natural field backgrounds, thereby improving sorghum yield estimation accuracy. It also contributes to increased sorghum production and harvest, as well as the enhancement of intelligent harvesting equipment for agricultural machinery.

## 1. Introduction

Sorghum, a cereal crop with a long history of cultivation in China, is often referred to as the “camel of grains” and “big and small rice” due to its resilience and versatility [1]. Its notable drought resistance and salt tolerance render it an essential ecological crop with diverse applications, including food, forage, and brewing, thereby playing a significant role in China’s economic development [2,3]. A critical factor in evaluating sorghum’s yield and quality is the sorghum spike. However, in practical sorghum production, yield measurement traditionally relies on manual counting or weighing post-harvest, which is both time-consuming and labor-intensive, prone to significant errors, and cannot be conducted continuously. The rapid advancement of deep learning technology in artificial intelligence offers a promising solution. By employing high-volume, high-precision rapid detection technology, it is now possible to predict yield data before the sorghum harvest, thereby enhancing sorghum cultivation practices. And it can detect sorghum ears in real time by carrying the embedded device on the harvester, calculate the loss rate during harvesting, and provide technical and data support for the improvement of intelligent harvesting equipment for agricultural machinery. It provides help for the development of sorghum harvesters to high quality, high efficiency, automation, and intelligence. Therefore, combining deep learning and image-processing technology to study a sorghum spike detection method with high detection accuracy and good detection effect is of great significance for the increase in sorghum production and harvesting and the innovation and improvement of intelligent agricultural machinery and equipment. In recent years, the rapid advancement of computer and artificial intelligence technologies has led to the widespread application of deep learning-based target detection methods in agriculture [4,5]. Researchers have employed various deep learning models, such as Faster Regional-Based Convolutional Neural Network (Faster RCNN) [6], You Only Look Once (YOLO) [7,8,9,10], Single Shot Multibox Detector (SSD) [11], and Mask Regional-Based Convolutional Neural Network (Mask RCNN) [12], for the recognition and detection of crop organs, including flowers [13,14], stems [15,16], leaves [17,18], and fruits [19,20]. These efforts have yielded significant results. For instance, based on the YOLOv8 model, by introducing the concept of shared convolutional layers and a visual transformer with a deformable attention mechanism, Ma proposed the YOLOv8-HD model [21], which significantly improved the detection accuracy of wheat seeds. Yang proposed an automatic tomato detection method based on the improved YOLOv8s model [22], which utilizes the depth-separable convolution (DSConv) technique instead of ordinary convolution to generate a large number of feature maps with a small number of computations, thus reducing the computational complexity. And the Dual Path Attention Gate Module (DPAG) is designed to improve the model’s detection accuracy in complex environments by enhancing the network’s ability to distinguish between tomatoes and background. Yang achieves a more efficient feature fusion network to improve the model’s detection performance for strawberries by introducing a residual network with learnable parameters and scaling normalization into the original residual structure of the Swin transformer of YOLOv8s [23]. Chen proposed an improved multitask deep convolutional neural network (DCNN) detection model based on YOLOv7 [24], which added two decoders to YOLOv7 for detecting tomato fruit clusters, fruit ripeness, and cluster ripeness. Subsequently, the loss function was designed according to the characteristics of multitasking, and scale-sensitive joint intersection (SIoU) was used instead of complete joint intersection (CIoU) to improve the recognition accuracy of the model. An improved yellow peach detection model (EMA-YOLO) based on YOLOv8 is proposed, which reduces the leakage and false detection rate of target small yellow peaches by introducing an EMA attention mechanism module to encode global information [25]. Incorporating an attention mechanism into the RCNN model enhances the feature extraction capability of the backbone network, enabling accurate segmentation of apple targets in complex backgrounds [26]. Additionally, the design of a structurally lightweight YOLOv5 model, with an added convolutional network to the improved backbone, effectively enhances the recognition of occluded apples by apple-picking robots in challenging orchard environments [27]. The integration of the Convolutional Block Attention Module (CBAM) and the a-1OU loss function into the YOLOv5 model has improved the recognition of citrus fruits in natural settings [28]. In cereal crops, Zhang et al. developed a wheat spike detection method utilizing an attention mechanism and pyramid network, significantly improving the detection of occluded and smaller spikes [29]. Furthermore, Zhang et al. introduced unfolded convolution in the Faster R-CNN model to optimize the Inception_ResNet-v2 feature extraction network, resulting in a rice spike detection model applicable at different growth stages [30]. Xiao et al. enhanced the YOLOv4 model with the CBAM attention mechanism, improving the network’s feature extraction from grapes by focusing attention on the CSPDarknet53 and PANet networks, thereby reducing interference from complex scenes and achieving high accuracy in grape detection [31]. Yang et al. employed the Faster R-CNN model and SegNet network for rice spike detection and segmentation, developing a method for extracting rice phenotypic features and predicting spike weight [32]. Zhao et al. proposed a wheat spike detection method based on an improved YOLOv5 model, effective under occlusion and overlapping conditions in UAV images [33].

In recent research on deep learning-based crop organ recognition, significant advancements have been made in improving detection accuracy and speed. However, these deep learning models typically require high-performance personal computer platforms, rendering them unsuitable for embedded devices with limited computing resources. To address this limitation and facilitate the application of deep learning models in real-world production, it is essential to reduce the models’ parameters and complexity. Two primary methods for achieving lightweight deep learning models are model compression and the design of lightweight model structures. For instance, researchers have suggested using lightweight models such as MobileNet v3 and MobileNet v2 to replace traditional deep learning models in backbone feature extraction networks [34,35,36]. Wu et al. developed a lightweight and enhanced YOLOv3 model for apple detection, which employs depth-separable convolution instead of standard convolution and incorporates a feature extraction network composed of multiple residual blocks in series. This model was successfully implemented on both a workstation and an Nvidia TX2-embedded development board to detect apples amidst complex fruit tree environments [37]. Additionally, Wu et al. introduced a channel pruning algorithm to optimize the YOLOv4 model by reducing its parameters, size, and inference time, thereby enabling the detection of apple blossoms in natural settings [38]. Yang et al. proposed a CenterNet model for rapid multi-apple target detection without anchor boxes, utilizing the lightweight Tiny Hourglass24 as the backbone network and optimizing the residual module to achieve efficient detection in dense scenes [39]. Despite these advancements, the lightweight nature of model structures can sometimes compromise detection accuracy, particularly in identifying occluded, adhered, or small-sized targets in complex environments. To mitigate this issue, numerous studies have incorporated attention mechanisms and multiscale detection strategies. For example, Li et al. developed a YOLOv4-tiny model for the fast and accurate detection of green peppers, employing attention mechanisms for multiscale detection and adaptive feature fusion to enhance both speed and performance [40]. Similarly, Wang et al. proposed an improved YOLOv4-tiny model for blueberry fruit detection, integrating the CBAM attention mechanism into the feature pyramid to maintain accuracy and speed in fruit recognition [41].

In summary, the research on crop detection based on target detection technology has already had certain foundations and achievements, but there are still some problems, such as backward research models, low model detection accuracy, and insufficient application. There is also no high-performance sorghum spike detection model proposed for the problems of different sorghum growth, serious occlusion between sorghum ears, and unclear boundary contours in natural field environments. Therefore, this study proposes an enhanced sorghum spike detection method based on the YOLOv8s model. This method introduces the Gold feature pyramid module to improve the information fusion capability of the neck module in the YOLOv8s model. Additionally, the SPPF module is refined using the LSKA attention mechanism to heighten the network’s focus on significant features. Furthermore, a loss function combining the Focal-EIOU loss is employed to address the class imbalance problem inherent in sorghum detection. The improved model developed in this study is optimized for mobile devices with low computational power, ensuring robust detection performance even in the complex natural field background.

## 2. Materials and Methods

### 2.1. Image Acquisition

The original images of sorghum spikes were collected from the Sorghum Practice Base of Shanxi Agricultural University. The collected varieties were Jinnuo 3 and Jinzai 22, and the collection time was from August to October 2023. As shown in Figure 1, in order to solve the occlusion problem of sorghum spikes in the natural field background and to realize the requirement of large-area recognition, a total of 2000 images of sorghum spikes at different positions and angles were collected by changing the shooting distance and shooting angle from the sorghum spike. The collected images include a flat shot taken parallel to the sorghum spike at a position one meter from the first row of sorghum spikes, a top shot taken vertically three meters from the ground, and a sorghum spike image taken at a tilted camera of 45 degrees at a height of three meters from the ground when simulating a harvester-mounted camera. The captured sorghum spike images had a resolution of 4032 pixels × 3024 pixels and were stored in jpg format.

There are numerous complications in sorghum spike images taken in natural field environments, such as heavy shading between sorghum spikes, dense distribution of sorghum spikes, and windy conditions that result in unclear captured images; therefore, the collected images need to be screened to remove those that are poorly captured and cannot be recognized and labeled. Considering the laboratory computer hardware and GPU performance, the image pixels are compressed to 1024 pixels × 768 pixels to speed up the model training. LabelImg (v1.8.1) is used to manually label the dataset according to the YOLO format, and the label file is saved in XML format. In order to alleviate the overfitting problem caused by the small dataset and improve the generalization ability of the model training results, Python is used to amplify the image data, and the sorghum dataset is rotated, flipped, mirrored, and brightness adjusted, as shown in Figure 2. After data enhancement of the original sorghum spike dataset, a total of 4500 images were obtained. The dataset contains pictures of the sorghum milk ripening period, wax ripening period, and full ripening period taken from different angles, and the dataset is divided into a training set, verification set, and test set according to the ratio of 8:1:1.

### 2.2. Construction of a Sorghum Spike Detection Model

#### 2.2.1. YOLOv8

Released by Ultralytics in January 2023, YOLOv8 [42] is a next-generation real-time target detection model that demonstrates enhanced feature extraction and target detection capabilities. This improvement is attributed to its advanced backbone network and neck architecture. The model employs an anchorless segmentation head design, which not only increases accuracy but also enhances the efficiency of the detection process. Furthermore, YOLOv8 is designed to maintain an optimal balance between accuracy and speed, rendering it suitable for a variety of real-time target detection tasks. These tasks encompass detection, segmentation, pose estimation, tracking, and classification across different application areas.

Figure 3 illustrates the structure of the YOLOv8 model, which is primarily composed of three components: the backbone, neck, and head. The model is available in five versions—YOLOv8n, YOLOv8s, YOLOv8m, YOLOv8l, and YOLOv8x—each differing in the number of feature extraction modules and convolutional kernels within the backbone [43]. These versions exhibit a progressive increase in weights and model volumes. For this study, which focuses on estimating sorghum yield in real-world environments, YOLOv8s was chosen as the foundational model. This selection was based on a balanced consideration of detection speed, accuracy, and model size, facilitating the model’s deployment on a mobile platform.

#### 2.2.2. Gold Feature Pyramid Module

The neck structure of YOLOv8s utilizes the traditional Feature Pyramid Network (FPN) architecture, which is designed for multiscale feature fusion. This architecture consists of multiple branches that allow for the fusion of features from neighboring levels. However, it can only fully integrate features from adjacent levels, while information from non-adjacent levels must be accessed indirectly through “recursive” methods. As illustrated in Figure 4, the traditional FPN arranges existing levels, 1, 2, and 3, from top to bottom, facilitating fusion between different levels. For instance, when level 1 requires information from level 2, it can directly acquire and integrate these data. Conversely, if level 1 needs information from level 3, it must first combine the information from levels 2 and 3 and then indirectly access level 3 data through level 2. This method addresses the challenge of information transfer inherent in the traditional FPN structure. In such structures, when cross-layer information fusion is necessary, the traditional FPN is unable to transmit information without loss, which limits the effectiveness of YOLOv8s in achieving optimal information fusion. To overcome this limitation, this study introduces a novel collection and distribution mechanism known as Gold. This mechanism significantly enhances the neck’s information fusion capability by globally integrating multiscale features and infusing global information into higher levels, thereby improving the model’s performance across different object sizes.

The Gold feature pyramid module introduces an innovative mechanism for the collection and distribution of information, based on the concept of global information fusion, to enhance the efficiency of information exchange in YOLO. By integrating multi-layer features globally and infusing this global information into higher levels, the module significantly improves the information fusion capability of the neck, thereby enhancing the model’s performance in detecting objects of varying sizes. The Gold mechanism comprises two distinct branches: a shallow collection and distribution branch and a deep collection and distribution branch. These branches utilize convolutional basis blocks and attention basis blocks to extract and fuse feature information effectively. The network structure of the Gold module is illustrated in Figure 5.

The algorithm is structured into two stages, Low-Gold and High-Gold. Low-Gold demonstrates a clear advantage in detecting small targets, whereas High-Gold is more adept at detecting large targets. Low-Gold and High-Gold consist of three key modules, respectively: Feature Alignment Module (FAM), Information Fusion Module (IFM), and Information Injection Module (Inject). The FAM is tasked with collecting feature information at various levels and standardizing the specifications. Following this, the IFM takes the unified feature information from the FAM and integrates it to generate global information for distribution. Subsequently, the Information Injection Module (Inject) completes the integration of feature information across different levels after the IFM has processed it. This module injects the fused information into various layers, akin to the Feature Pyramid Network (FPN), thereby ultimately enhancing the feature detection capabilities of these layers. Figure 6 illustrates the structural flow charts of Low-Gold and High-Gold.

Low-Gold is mainly used to fuse the shallow feature information of the model, and its process is divided into three steps: alignment, fusion, and injection. Firstly, the features of different scales are aligned with the LowFAM module, and *B2*, *B3*, *B4*, and *B5* are unified to the size of [*h*/4,*w*/4] of *B4*, which is concatenated on channel; the output of Low-FAM is extracted by multi-layer RepBlock, which is divided into two parts: *inject P3* and *inject P4* as the input of inject; finally, the Inject module injects the information by calculating the formula as follows, where *global* represents the features that are fused together in multiple layers using the FAM module, and *local* represents the features in each layer.
(1)$$F_{align}=Low-FAM([B2,B3,B4,B5])$$


(2)
$$F_{fuse}=RepBlock(F_{align})$$



(3)
$$F_{inject\_P3},F_{inject\_P4}\;=\;Split(F_{fuse})$$



(4)
$$F_{global\_act\_Pi}=resize(Sigmoid(Conv_{act}(F_{inject\_Pi})))$$



(5)
$$F_{global\_embed\_Pi}\;=\;resize({Conv}_{global\_embed\_Pi}(F_{inject\_Pi}))$$



(6)
$$F_{att\_fuse\_Pi}\;=\;Conv_{local\_embed\_Pi}(Bi)\;\times \;F_{global\_act\_Pi}\;+\;F_{global\_embad\_Pi}$$



(7)
$$Pi\;=\;RepBlock(F_{att\_fuse\_Pi})$$


High-Gold is similar to Low-Gold in structure, and only *P3*, *P4*, and *P5* are used for input. Since the feature size of the High-FAM output is smaller, in order to fully integrate the global information, the Repblock in Low-Gold is replaced by a transformer module based on convolution, the linear linear layer is replaced by conv, and the layer norm is replaced by batch norm. Finally, the injection information is calculated by the Inject module, and the calculation formula is as follows:(8)$$F_{align}=High-FAM([P3,P4,P5])$$
(9)$$F_{fuse}=Transformer(F_{align})$$
(10)$$F_{inject\_N4},F_{inject\_N5}\;=\;Split(Conv1\;\times \;1(F_{fuse}))$$
(11)$$F_{global\_act\_Ni}=resize(Sigmoid(Conv_{act}(F_{inject\_Ni})))$$
(12)$$F_{global\_embed\_Ni}=resize({Conv}_{global\_embed\_Ni}(F_{inject\_Ni}))$$
(13)$$F_{att\_fuse\_Ni}=Conv_{local\_embed\_Ni}(Pi)\times F_{global\_act\_Ni}+F_{global\_embad\_Ni}$$
(14)$$Ni=RepBlock(F_{att\_fuse\_Ni})$$

#### 2.2.3. LSKA Attention Mechanisms

The Spatial Pyramid Pooling Layer (SPPF) in YOLOV8 is designed to effectively aggregate features at multiple scales. The integration of the Large and Separable Kernel Attention (LSKA) mechanism significantly enhances the network’s focus on crucial features, thereby improving overall model performance. This enhancement is achieved through the use of large and separable convolutional kernels, along with spatially expanding convolutions, which together capture extensive contextual information from an image. By generating an attention graph, the LSKA mechanism weights the original features according to this graph, thereby refining the feature aggregation process. Consequently, the LSKA attention mechanism not only bolsters the SPPF module’s capacity to aggregate multiscale features but also reduces computational complexity and memory usage.

The structure of the LSKA model can be categorized into three main components: the initialization convolutional layer, the spatial expansion convolutional layer, and the fusion and application of attention. Firstly, the initialization convolutional layer is responsible for extracting features from the horizontal and vertical directions of the input feature map to generate an initial attention map that helps the model to focus on the important parts of the image. The obtained attention map is then further extracted by spatially expanding convolutional layers using spatially expanding convolutions with different expansion rates. These convolutional layers are able to cover a larger receptive field and capture a wider range of contextual information without increasing the computational cost. By operating horizontally and vertically, they allow for more detailed processing of image features and enhance the model’s understanding of spatial relationships in the image. Finally, after a series of convolutional operations, LSKA fuses the obtained features through the last convolutional layer (conv1) to generate the final attention map. This attention map is subjected to an element-level multiplication operation with the original input feature map (u), so that each element in the original feature map is weighted according to the value of the attention map, highlighting important features and suppressing unimportant ones. In summary, the LSKA attention mechanism captures the contextual information of an image by utilizing large and separable convolutional kernels as well as spatially expanding convolutions and generates an attention graph. The original features are then weighted by this attention map as a way to enhance the network’s attention to important features and improve the performance of the model. In this study, the improvement process is depicted in Figure 7, illustrating the integration of the LSKA module within the SPPF structure. Specifically, the LSKA module is inserted between all the maximum pooling layers (MaxPool2d) and the second convolutional layer (Conv). In the forward method of the code, the input x initially passes through the first convolutional layer, cv1. Subsequently, it traverses the three maximal pooling layers, denoted as *m*. The outputs from these layers are then concatenated using the Concat function. This concatenated output is subsequently processed by the LSKA module. Following the processing by the LSKA module, the outputs are directed into the second convolutional layer, cv2.

#### 2.2.4. Focal-EIOU Loss

In the target detection model, the loss function primarily comprises three components: bounding box loss, classification loss, and object confidence loss. In the YOLOv8s model, the Complete Intersection over Union (CIOU) loss is employed to calculate the bounding box loss. CIOU loss enhances the Distance Intersection over Union (DIOU) loss by incorporating an aspect ratio term, which aims to align the predicted bounding box more closely with the actual box. However, this aspect ratio term does not directly address the true differences between the width and height, nor does it adequately consider the confidence level. Consequently, it fails to provide sufficient gradient signals to guide the network in accurately adjusting the bounding box dimensions. Furthermore, CIOU loss is computationally complex, demanding more resources during training, and it does not entirely resolve all issues associated with bounding box regression, occasionally resulting in significant errors. Additionally, the use of CIOU loss can lead to an unstable training process, necessitating meticulous hyperparameter tuning.

In predicting target bounding box regression, the insufficiency of CIOU loss and the issue of unbalanced training samples present significant challenges. Specifically, there is a disparity in the number of high-quality anchor frames, which have small regression errors, compared to low-quality anchor frames, which exhibit large errors within a single image. This imbalance results in poorer quality anchor frames generating excessive gradients, thereby adversely affecting the training process. To address these issues, this study employs Focal-EIOU loss in conjunction with EIOU loss. This approach aims to differentiate high-quality anchor frames from low-quality ones by considering the gradient, as demonstrated in the following formula:(15)$$L_{EIOU}=L_{IOU}+L_{dis}+L_{asp}=1-IOU+\frac{\rho^{2}(b,b^{gt})}{{\left(w^{c}\right)}^{2}+{\left(h^{c}\right)}^{2}}+\frac{\rho^{2}(w,w^{gt})}{{\left(w^{c}\right)}^{2}}+\frac{\rho^{2}(h,h^{gt})}{{\left(h^{c}\right)}^{2}}$$
(16)$$IOU=\frac{A\cap B}{A\cup B}$$
(17)$$L_{Focal-EIOU}=IOU^{\gamma }L_{EIOU}$$

Figure 8 illustrates the parameters involved in the calculation; *w^c^* and *h^c^* are the width and height of the minimum circumscribed rectangle of the anchor frame and the target frame. *ρ* is the Euclidean distance between *b* and *b^gt^*, and *γ* is the parameter to control the suppression degree of outliers. The Focal-EIOU loss function builds upon the Complete Intersection over Union (CIOU) by calculating the difference in width and height, rather than the aspect ratio. This approach directly minimizes the discrepancy between the predicted frame box and the actual frame in terms of width and height, thereby accelerating the convergence speed. Additionally, Focal-EIOU demonstrates superior performance in handling challenging samples, which enhances the robustness and accuracy of the target detection algorithm.

### 2.3. Improved Modeling

The improved YOLOv8s model depicted in Figure 9 is designed for real-time detection of sorghum spikes in natural field conditions. To enhance the model’s performance in detecting objects of varying sizes, the information fusion capability of the neck module is augmented by integrating the Gold feature pyramid module. Additionally, the SPPF module is refined through the incorporation of the LSKA attention mechanism, which heightens the network’s focus on critical features, thereby improving overall model performance. To address the issue of class imbalance in sorghum detection, a loss function combining Focal-EIOU loss is employed. Consequently, the sorghum spike detection model, YOLOv8s-Gold-LSKA, is developed, leveraging both the Gold feature pyramid module and the LSKA attention mechanism.

## 3. Results and Discussion

In this study, the improved YOLOv8s network was trained using a stochastic gradient descent (SGD) approach in an end-to-end manner. The training process involved 300 iterations, with a batch size of four samples per iteration. Regularization was applied at each step through the Batch Normalization (BN) layer, ensuring the continuous update of the model’s weights. The specific configuration parameters of the operating environment are detailed in Table 1. As depicted in Figure 10, the loss values of the model significantly decreased during the initial 200 rounds of training, with the model achieving convergence around the 250th round. Consequently, the model output after 300 rounds of training was designated as the sorghum spike detection model for this study.

### 3.1. Evaluation of Indicators

In this study, precision, recall, mean average precision (mAP), and F1-score are used to evaluate the detection performance of the model. The specific calculation methods are as follows:(18)$$Precision=\frac{TP}{TP+FP}\times 100\%$$
(19)$$Recall=\frac{TP}{TP+FN}\times 100\%$$
(20)$$mAP=\frac{1}{C}\sum_{M=i}^{N}P(k)\Delta R(k)\times 100\%$$
(21)$$F_{1}\text{-}score=\frac{2\times Precision\times Recall}{Precision+Recall}$$

In this context, TP represents the number of correctly identified sorghum spikes, FP denotes the number of falsely identified or unrecognized spikes, and FN refers to the number of incorrectly identified sorghum spikes. The variable C indicates the number of sorghum spike categories. M and N represent the number of Intersection over Union (IoU) thresholds, respectively. Furthermore, P(k) and R(k) signify the precision and recall rates, respectively.

The F1 score is the harmonic mean of precision and recall. Network complexity is measured using parameters and floating-point operations (FLOPs), with smaller values indicating lower complexity.

### 3.2. Ablation Experiments

To assess the effectiveness of the improved model developed in this study, ablation experiments were conducted utilizing a self-constructed sorghum spike dataset. The results of these experiments are detailed in Table 2. In this table, a “√” symbol signifies that the corresponding method was employed to enhance the model, whereas a “-” symbol indicates that the method was not utilized.

The integration of the Gold module into the YOLOv8s model resulted in an increase in the recall value from 65.53% to 69.73% and an enhancement of the mAP value from 79.23% to 81.58%. These findings suggest that the Gold module enhances the information fusion capability of the backbone network, thereby improving the model’s performance in detecting sorghum ears of varying sizes. Similarly, the introduction of the LSKA module alone led to a 1.32% increase in the recall value and a reduction in the model size by 0.79 MB. This indicates that the LSKA module enhances the SPPF module’s ability to aggregate features across multiple scales, reduces computational complexity and memory usage, and consequently improves overall model performance. Initially, the YOLOv model had a size of 9.61 MB; however, the incorporation of Focal-EIOU loss reduced the model size to 7.74 MB, along with a decrease in the number of parameters and floating-point operations. This demonstrates that the Focal-EIOU loss contributes to a lightweight effect on the YOLOv8s model. Although the Focal-EIOU loss primarily aims to minimize the loss of the bounding box during the regression process and accelerate model convergence, it only marginally enhances detection performance when used alone. Nevertheless, when the LSKA module and Focal-EIOU loss are applied together, the model’s performance is comprehensively improved across all detection metrics. This combined application underscores the synergistic effect of the LSKA module and Focal-EIOU loss in optimizing the YOLOv8s model.

The F1 value and mAP value of the improved model in this study were 81.19% and 85.86%, respectively, which were 5.42% and 6.63% higher than the original YOLOv8s model. The model size is reduced from 9.61 MB to 7.48 MB, a total reduction of 2.13 MB. The parameter quantity and floating-point operation are also significantly reduced. The results show that the improved YOLOv8s-Gold-LSKA model has obviously improved the detection indicators except the detection time, and the model complexity is significantly reduced.

Figure 11 shows the visualization of the YOLOv8s-Gold-LSKA model for sorghum detection under different weather and maturity periods. The image captured from the front demonstrates a relatively simple background and high recognition accuracy. However, it presents an occlusion issue, as the sorghum spikes overlap, making it challenging to detect and recognize the spikes positioned at the back. Conversely, in the image taken at a 45-degree angle, the occlusion between sorghum spikes is reduced, resulting in higher recognition accuracy for the spikes in the foreground. Nevertheless, during the ripening period, the sorghum spikes exhibit very similar colors, which complicates detection. As the shooting distance increases, the actual angle for capturing distant sorghum spikes decreases, leading to inconsistencies in size and indistinct border outlines in the image. These factors collectively increase the difficulty of recognition and subsequently reduce the overall recognition accuracy.

The accuracy of detection based on pictures is not high due to severe occlusion between sorghum spikes in images taken from the front. Therefore, detection can be performed based on images taken from directly above, avoiding the occlusion problem and providing better detection results. In the estimation of actual sorghum yield, the process involves recognizing the number of sorghum spikes within a unit area, multiplying this by the total area, and then further multiplying by the average quality of the sorghum. This method allows for an accurate yield estimation. The improved model demonstrates enhanced detection capabilities for sorghum spikes, particularly those photographed from close and frontal perspectives within a natural field background. Consequently, it effectively accomplishes the task of detecting sorghum spikes in their natural field environment.

### 3.3. Comparison of Model Performance

Under the same experimental environment, the improved YOLOv8s-Gold-LSKA model in this study improved the F_1_ and mAP values by 5.42% and 6.63%, respectively, and the model size was reduced by 2.13 MB, as well as all other parameters, when compared with the base model YOLOv8s. Compared withYOLOv5s, SSD, and YOLOv8, tested on a self-constructed sorghum spike dataset, the comparative results, as presented in Table 3, demonstrate that the F1 scores for the YOLOv5s, SSD, and YOLOv8 models were 71.08%, 73.65%, and 75.77%, respectively. In contrast, the improved YOLOv8s-Gold-LSKA model achieved an F1 score of 75.77%, representing increases of 10.11%, 7.54%, and 5.42% over the aforementioned models. Furthermore, the mean average precision (mAP) of the improved model was 85.86%, surpassing the mAP values of the YOLOv5s, SSD, and YOLOv8 models. Additionally, the detection time per image for the improved YOLOv8s-Gold-LSKA model was shorter than that of YOLOv5s and SSD, and its model size was smaller compared to YOLOv5s, SSD, and YOLOv8. These findings indicate that the improved YOLOv8s-Gold-LSKA model not only ensures superior detection performance but also offers enhanced detection speed and reduced model size. Consequently, the improved YOLOv8s-Gold-LSKA model is more suitable for sorghum spike detection tasks in natural field environments.

## 4. Conclusions

This paper presents an enhanced sorghum spike detection model, based on YOLOv8s, designed for recognizing sorghum spikes in natural field environments. The model incorporates the Gold feature pyramid module and the LSKA attention mechanism and Focal-EIOU loss function to address the challenge of detecting difficult targets in such contexts. The improved YOLOv8s-Gold-LSKA model achieves a precision value of 90.72%, a recall value of 76.81%, an F1 score of 81.19%, a mean average precision (mAP) of 85.86%, a model size of 7.48 MB, and an average detection time of 0.0168 s per image. Compared with the basic model YOLOv8s, the F_1_ value and mAP value were increased by 5.42% and 6.63%, respectively, the model size was reduced by 2.13 MB, and other parameters were also improved. Experimental results demonstrate that this model effectively enhances the detection of challenging samples while maintaining a lightweight model size and rapid detection speed. When compared to YOLOv5s, SSD, YOLOv8, and the original YOLOv8s model, the YOLOv8s-Gold-LSKA model exhibits superior performance across various evaluation metrics, including detection accuracy, parameter count, detection time, and visualization of detection results.

These findings underscore the effectiveness of the improvements made in this study and suggest that the YOLOv8s-Gold-LSKA model holds potential application value for detection tasks in natural field settings. Consequently, this study can provide help for sorghum cultivation by predicting yield data before sorghum harvest, provide new ideas and technical support for sorghum spike detection and yield estimation, and promote the realization of increased sorghum production and income. And it can detect sorghum spikes in real time by carrying embedded equipment on the harvester, calculate the loss rate during harvesting, and provide help for the development of sorghum harvesters to high quality, high efficiency, automation, and intelligence. It is of great significance to the innovation and improvement of intelligent agricultural equipment and has a positive impact on the scientific and intelligent agricultural production activities of agricultural workers.

## Figures and Tables

**Figure 1 sensors-25-00104-f001:**
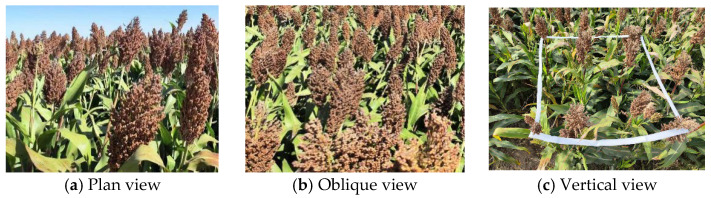
Images of sorghum spikes taken at different angles.

**Figure 2 sensors-25-00104-f002:**
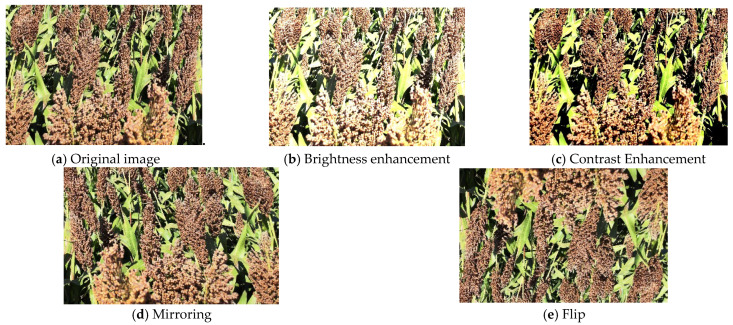
Data enhancement.

**Figure 3 sensors-25-00104-f003:**
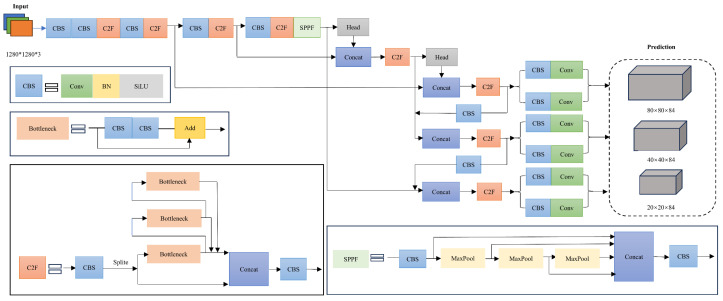
Network structure of YOLOv8s.

**Figure 4 sensors-25-00104-f004:**
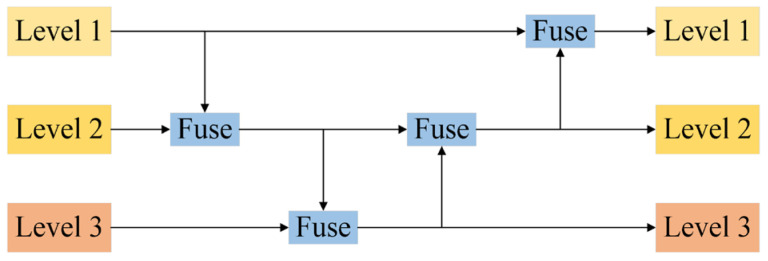
Traditional neck structure.

**Figure 5 sensors-25-00104-f005:**
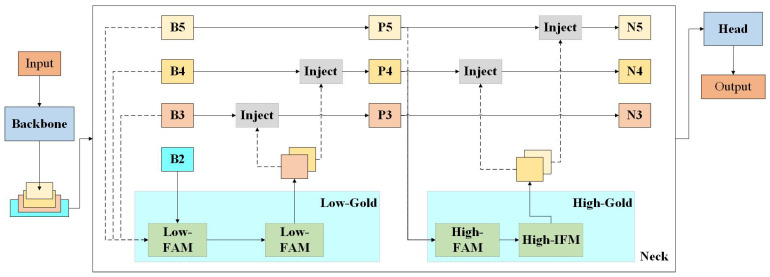
Network structure of Gold.

**Figure 6 sensors-25-00104-f006:**
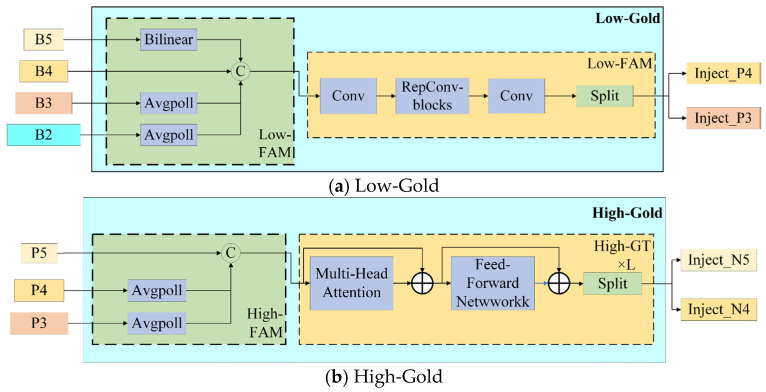
Flowchart of the structure of Low-Gold and High-Gold.

**Figure 7 sensors-25-00104-f007:**
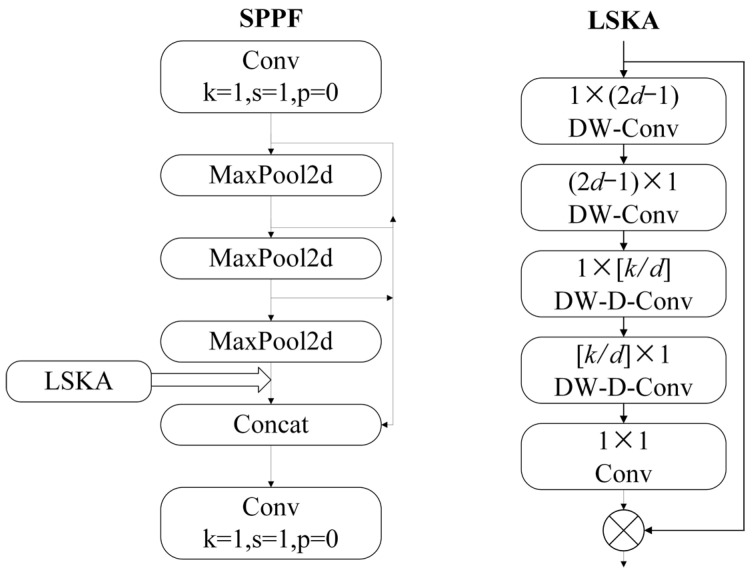
Module structure diagram.

**Figure 8 sensors-25-00104-f008:**
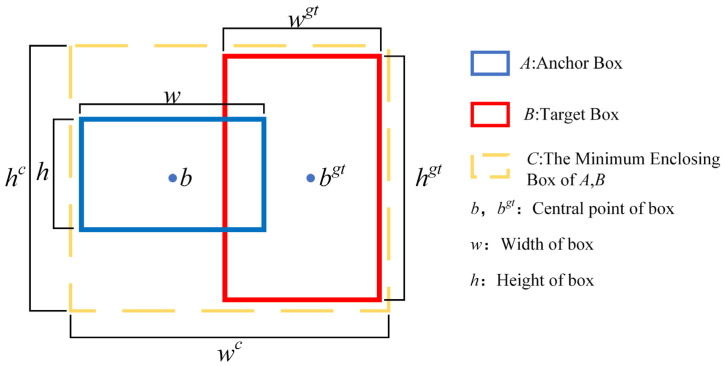
Schematic diagram of each variable.

**Figure 9 sensors-25-00104-f009:**
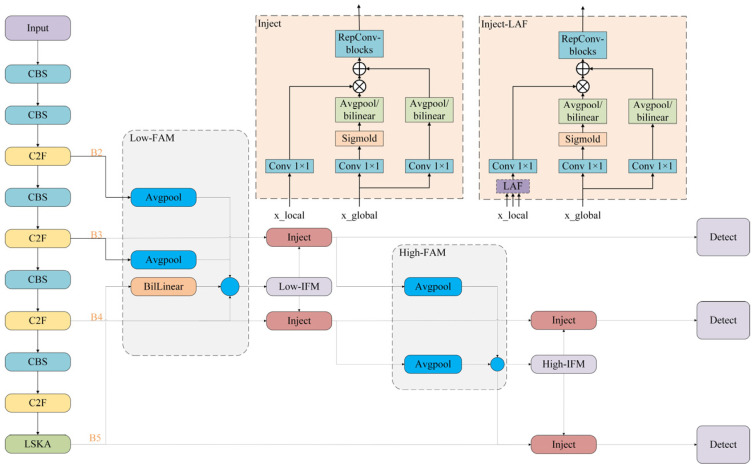
Improved YOLOv8s-Gold-LSKA model.

**Figure 10 sensors-25-00104-f010:**
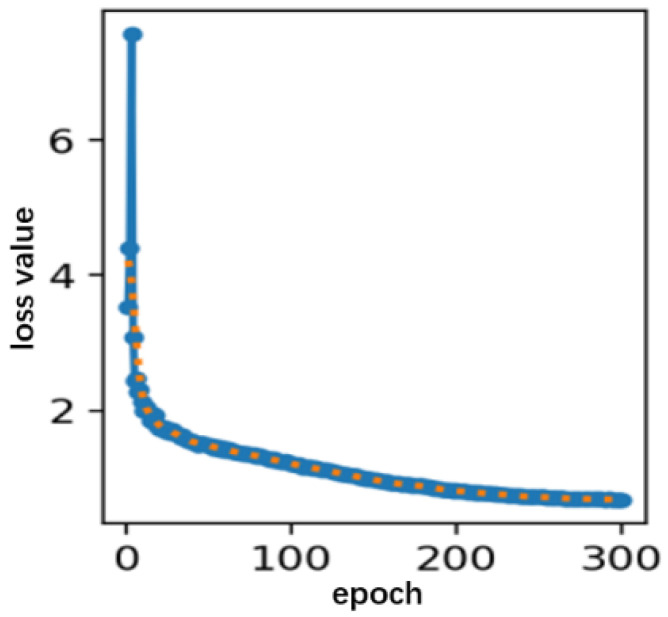
Training loss curve of the improved model.

**Figure 11 sensors-25-00104-f011:**
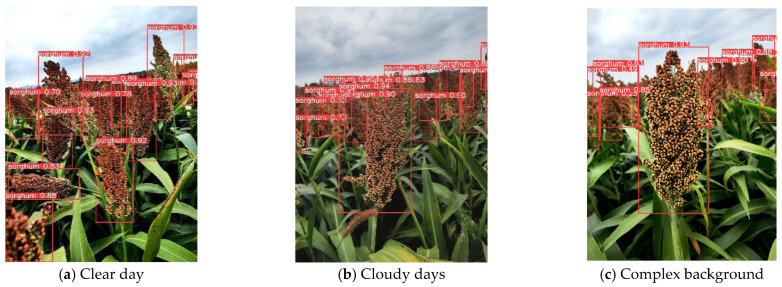
Visualization of YOLOv8s-Gold-LSKA model detection results.

**Table 1 sensors-25-00104-t001:** Operating environment configuration parameters.

Designation	Releases
Operating system	Windows 10 (x64)
CPU	AMD Ryzen 7 5800H
GPU	NVIDIA GeForce RTX3060 latop
RAM	32 GB
PyCharm	2022.2
CUDA	11.4
CuDNN	8.2.4
Python	3.8.5

**Table 2 sensors-25-00104-t002:** Results of ablation experiments.

Gold	LSKA	Focal-EIOU Loss	P/%	R/%	F_1_/%	mAP/%	Average Detection Time per Image/s	Model Size/MB
-	-	-	89.81	65.53	75.77	79.23	0.0165	9.61
√	-	-	90.19	69.73	78.65	81.58	0.0181	9.43
-	√	-	89.24	66.85	76.50	79.36	0.0158	8.82
-	-	√	87.40	69.91	77.68	78.29	0.0165	7.74
√	√	-	90.65	70.96	79.61	82.52	0.0180	8.47
√	-	√	91.14	75.23	82.42	85.06	0.0159	8.06
-	√	√	90.45	73.10	80.86	83.51	0.0172	7.79
√	√	√	90.71	76.81	81.19	85.86	0.0168	7.48

**Table 3 sensors-25-00104-t003:** Comparison of detection performance of different network models.

Model	P/%	R/%	F_1_/%	mAP/%	Average Detection Time per Image/s	Model Size/MB
YOLOv5s	86.22	60.47	71.08	73.08	0.0183	10.45
SSD	88.35	63.14	73.65	77.74	0.0187	11.26
YOLOv8	89.81	65.53	75.77	79.23	0.0165	9.61
YOLOv8s-Gold-LSKA	90.72	76.81	81.19	85.86	0.0168	7.48

## Data Availability

Data are contained within the article.

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
