# Peer review of "Sorghum Spike Detection Method Based on Gold Feature Pyramid Module and Improved YOLOv8s"

_sensors, 2024, doi:10.3390/s25010104_

Round 1
Reviewer 1 Report
Comments and Suggestions for Authors
The sorghum spike detection method is very important for its yield estimation.
The questions of the paper are as follows.
1) The authors should provide successful scenarios in which the YOLOv8s model has been applied in the introduction.
2) The author should explain how to quantify the angle required for photography during the study, as shown in Figure 1. However, it may be difficult to grasp the angle of taking photos when there is wind.
3) “YOLOv8s-GOLD-LSKA model achieves a Precision value of 90.72%”. However, The author should discuss the interfering factors that affect the accuracy of the model, such as weather. At the same time, it should be explained how to remove these noise signals that may exist in the application.
Author Response
For research article
Response to Reviewer 2 Comments
|
||
1. Summary |
|
|
Thank you very much for taking the time to review this manuscript. Please find the detailed responses below and the corresponding revisions/corrections highlighted/in track changes in the re-submitted files. [This is only a recommended summary. Please feel free to adjust it. We do suggest maintaining a neutral tone and thanking the reviewers for their contribution although the comments may be negative or off-target. If you disagree with the reviewer's comments please include any concerns you may have in the letter to the Academic Editor.]
|
||
2. Questions for General Evaluation |
Reviewer’s Evaluation |
Response and Revisions |
Does the introduction provide sufficient background and include all relevant references? |
Can be improved |
Changes have been made in the manuscript |
Is the research design appropriate? |
Can be improved |
Changes have been made in the manuscript |
Are the methods adequately described? |
Can be improved |
Changes have been made in the manuscript |
Are the results clearly presented? |
Can be improved |
Changes have been made in the manuscript |
Are the conclusions supported by the results? |
Can be improved |
Changes have been made in the manuscript |
3. Point-by-point response to Comments and Suggestions for Authors |
||
Comments 1: The authors should provide successful scenarios in which the YOLOv8s model has been applied in the introduction. |
||
Response 1: Thank you for pointing this out. In the introduction, the scenarios of successful application of YOLOv8 and YOLOv8s are supplemented. The details of the modifications and additions are set out below. For more information, see lines 61 to 73. For instance, based on the YOLOv8 model, by introducing the concept of shared convolutional layers and a visual transformer with a deformable attention mechanism, Ma proposed the YOLOv8-HD model [21], which significantly improved the detection accuracy of wheat seeds. Yang proposed an automatic tomato detection method based on the improved YOLOv8s model [22], which utilizes the depth-separable convolution (DSConv) technique instead of ordinary convolution to generate a large number of feature maps with a small number of computations, thus reducing the computational complexity. And the Dual Path Attention Gate Module (DPAG) is designed to improve the model's detection accuracy in complex environments by enhancing the network's ability to distinguish between tomato and background. Yang achieves a more efficient feature fusion network to improve the model's detection performance for strawberries by introducing a residual network with learnable parameters and scaling normalization into the original residual structure of the Swin transformer of YOLOv8s [23]. |
||
Comments 2: The author should explain how to quantify the angle required for photography during the study, as shown in Figure 1. However, it may be difficult to grasp the angle of taking photos when there is wind. |
||
Response 2: Thank you for pointing this out. Thank you for pointing this out. I've added a description of quantifying metrics such as shooting distance and height when acquiring images, and added a description of reducing interference from factors such as windy conditions by filtering the dataset. The details of the modifications and additions are set out below. For more information, see lines 155 to 159, 171 to 175. The collected images include a flat shot taken parallel to the sorghum spike at a position one meter from the first row of sorghum spikes, a top shot taken vertically three meters from the ground, and a sorghum spike image taken at a tilted camera of 45 degrees at a height of three meters from the ground when simulating a harvester-mounted camera. There are numerous complications in sorghum spike images taken in natural field environments such as heavy shading between sorghum spikes, dense distribution of sorghum spikes, and windy conditions that result in unclear captured images; therefore, the collected images need to be screened to remove those that are poorly captured and cannot be recognized and labeled. |
||
Comments 3: "YOLOv8s-GOLD-LSKA model achieves a Precision value of 90.72%". However, The author should discuss the interfering factors that affect the accuracy of the model, such as weather. At the same time, it should be explained how to remove these noise signals that may exist in the application. |
||
Response 3: Thank you for pointing this out. With regard to confounding factors affecting the accuracy of the model, a description of the reduction of confounding factors through dataset screening and data enhancement is added in Section 2.1, and a description of the reduction of sorghum spikes occluding each other by varying the angle of the shot is added to the conclusions. The details of the modifications and additions are set out below. For more information, see lines 171 to 175, 529 to 534. |
There are numerous complications in sorghum spike images taken in natural field environments such as heavy shading between sorghum spikes, dense distribution of sorghum spikes, and windy conditions that result in unclear captured images; therefore, the collected images need to be screened to remove those that are poorly captured and cannot be recognized and labeled.
In the natural field environment, sorghum spikes are clearer and more complete in images taken from the front, and can be detected in real time based on video to estimate yields, but due to the serious occlusion between the sorghum spikes, the accuracy of detection based on pictures is not high. In picture detection, detection can be based on images taken from directly above to avoid the occlusion problem, and the detection effect is good.
Reviewer 2 Report
Comments and Suggestions for Authors
This study proposes an improved sorghum ear detection method based on the YOLOv8s model. The method enhances the information fusion capability of the neck module in the YOLOv8 model by integrating the GOLD feature pyramid module. Additionally, the SPPF module is improved with the LSKA attention mechanism to enhance the focus on key features. To address the class imbalance issue in sorghum detection and accelerate model convergence, a loss function combining Focal-EIOU is used. The improved YOLOv8s-GOLD-LSKA model significantly improves the accuracy of sorghum ear detection in natural field environments, achieving an accuracy of 90.72%, recall of 76.81%, mean average precision (mAP) of 85.86%, and an F1 score of 81.19%. This advancement provides technical support for the rapid and accurate identification of multiple sorghum ear targets, improving the accuracy of sorghum yield estimation and contributing to increased sorghum production and harvest. I believe the research in this paper is highly significant, but there are some areas for improvement. I think the following major changes are necessary:
1. In the introduction section, the importance of sorghum ear detection and the limitations of existing methods could be described in more detail to highlight the significance and innovation of this study.
2 In the methods section, the principles and implementation details of the GOLD feature pyramid module and LSKA attention mechanism could be explained more clearly, so readers can better understand these innovations.
3. In the discussion section, a more in-depth analysis of the model's performance in challenging scenarios, such as occlusion and color similarity, could be provided to offer more insights for further improving detection performance.
4. In the conclusion section, the innovative aspects and practical application value of this research could be emphasized more, offering valuable technical support for the development of smart agricultural equipment.
5. In the references section, more recent literature related to this study could be added to reflect a comprehensive understanding of the current research status in the field. Additionally, it is recommended to review and include recent YOLOv8-related literature, such as "Efficient Optimized YOLOv8 Model with Extended Vision" and "A Lightweight Strip Steel Surface Defect Detection Network Based on Improved YOLOv8," etc.
6. Image enhancement should be applied to the figures in the paper to improve image clarity, especially for Figures 3, 8, and 9.
7. The versions of YOLOv5 and YOLOv8 used for comparison in the paper are not specified, which may lead to ambiguity. It is suggested to provide the specific versions.
8. Please clarify whether you plan to make your code publicly available. If it is released, it would be very helpful to interested readers.
Author Response
For research article
Response to Reviewer 3 Comments
|
||
1. Summary |
|
|
Thank you very much for taking the time to review this manuscript. Please find the detailed responses below and the corresponding revisions/corrections highlighted/in track changes in the re-submitted files. [This is only a recommended summary. Please feel free to adjust it. We do suggest maintaining a neutral tone and thanking the reviewers for their contribution although the comments may be negative or off-target. If you disagree with the reviewer's comments please include any concerns you may have in the letter to the Academic Editor.]
|
||
2. Questions for General Evaluation |
Reviewer’s Evaluation |
Response and Revisions |
Does the introduction provide sufficient background and include all relevant references? |
Can be improved |
Changes have been made in the manuscript |
Is the research design appropriate? |
Yes |
|
Are the methods adequately described? |
Can be improved |
Changes have been made in the manuscript |
Are the results clearly presented? |
Can be improved |
Changes have been made in the manuscript |
Are the conclusions supported by the results? |
Can be improved |
Changes have been made in the manuscript |
3. Point-by-point response to Comments and Suggestions for Authors |
||
Comments 1: In the introduction section, the importance of sorghum ear detection and the limitations of existing methods could be described in more detail to highlight the significance and innovation of this study. |
||
Response 1: Thank you for pointing this out. In the introduction part, the description of the importance of sorghum detection and the limitations of existing methods is supplemented. The details of the modifications and additions are set out below. For more information, see lines 36 to 46, 133 to 139. However, in practical sorghum production, yield measurement traditionally relies on manual counting or weighing post-harvest, which is both time-consuming and labor-intensive, prone to significant errors, and cannot be conducted continuously. The rapid advancement of deep learning technology in artificial intelligence offers a promising solution. By employing high-volume, high-precision rapid detection technology, it is now possible to predict yield data before the sorghum harvest, thereby enhancing sorghum cultivation practices. And it can detect sorghum ears in real time by carrying the embedded device on the harvester, calculate the loss rate during harvesting, and provide technical and data support for the improvement of intelligent harvesting equipment for agricultural machinery. It provides help for the development of sorghum harvester to high quality, high efficiency, automation and intelligence. In summary, the research on crop detection based on target detection technology has already had certain foundation and achievements, but there are still some problems such as backward research model, low model detection accuracy and insufficient application. There is also no high-performance sorghum spike detection model proposed for the problems of different sorghum growth, serious occlusion between sorghum ears, and unclear boundary contours in natural field environments. Therefore, this study proposes an enhanced sorghum spike detection method based on the YOLOv8s model. |
||
Comments 2: In the methods section, the principles and implementation details of the GOLD feature pyramid module and LSKA attention mechanism could be explained more clearly, so readers can better understand these innovations. |
||
Response 2: Thank you for pointing this out. In the Methods section, the principles and details of the GOLD feature pyramid module and the LSKA attention mechanism are added. For more information, see lines 283 to 299, 312 to 330. |
||
Comments 3: In the discussion section, a more in-depth analysis of the model's performance in challenging scenarios, such as occlusion and color similarity, could be provided to offer more insights for further improving detection performance. |
||
Response 3: Thank you for pointing this out. With regard to confounding factors affecting the accuracy of the model, a description of the reduction of confounding factors through dataset screening and data enhancement is added in Section 2.1, and a description of the reduction of sorghum spikes occluding each other by varying the angle of the shot is added to the conclusions. The details of the modifications and additions are set out below. For more information, see lines 171 to 175, 529 to 534. |
||
There are numerous complications in sorghum spike images taken in natural field environments such as heavy shading between sorghum spikes, dense distribution of sorghum spikes, and windy conditions that result in unclear captured images; therefore, the collected images need to be screened to remove those that are poorly captured and cannot be recognized and labeled. In the natural field environment, sorghum spikes are clearer and more complete in images taken from the front, and can be detected in real time based on video to estimate yields, but due to the serious occlusion between the sorghum spikes, the accuracy of detection based on pictures is not high. In picture detection, detection can be based on images taken from directly above to avoid the occlusion problem, and the detection effect is good. |
||
Response 4: Thank you for pointing this out. A description of the innovative and practical application value of the study is supplemented in the conclusion section. The details of the modifications and additions are set out below. For more information, see lines 536 to 545. Consequently, this study can provide help for sorghum cultivation by predicting yield data before sorghum harvest, provide new ideas and technical support for sorghum spike detection and yield estimation, and promote the realization of increased sorghum production and income. And it can detect sorghum spikes in real time by carrying embedded equipment on the harvester, calculate the loss rate during harvesting, and provide help for the development of sorghum harvester to high quality, high efficiency, automation and intelligence. It is of great significance to the innovation and improvement of intelligent agricultural equipment, and has a positive impact on the scientific and intelligent agricultural production activities of agricultural workers. |
||
Comments 5: In the references section, more recent literature related to this study could be added to reflect a comprehensive understanding of the current research status in the field. Additionally, it is recommended to review and include recent YOLOv8-related literature, such as "Efficient Optimized YOLOv8 Model with Extended Vision" and "A Lightweight Strip Steel Surface Defect Detection Network Based on Improved YOLOv8," etc. |
||
Response 5: Thank you for pointing this out. I have carefully understood the two fields of ' Efficient Optimized YOLOv8 Model with Extended Vision ' and ' A Lightweight Strip Steel Surface Defect Detection Network Based on Improved YOLOv8 '. In the introduction, the successful cases of applying YOLOv8 and the papers related to ' Efficient Optimized YOLOv8 Model with Extended Vision ' are added. The details of the modifications and additions are set out below. For more information, see lines 61 to 82. For instance, based on the YOLOv8 model, by introducing the concept of shared convolutional layers and a visual transformer with a deformable attention mechanism, Ma proposed the YOLOv8-HD model [21], which significantly improved the detection accuracy of wheat seeds. Yang proposed an automatic tomato detection method based on the improved YOLOv8s model [22], which utilizes the depth-separable convolution (DSConv) technique instead of ordinary convolution to generate a large number of feature maps with a small number of computations, thus reducing the computational complexity. And the Dual Path Attention Gate Module (DPAG) is designed to improve the model's detection accuracy in complex environments by enhancing the network's ability to distinguish between tomato and background. Yang achieves a more efficient feature fusion network to improve the model's detection performance for strawberries by introducing a residual network with learnable parameters and scaling normalization into the original residual structure of the Swin transformer of YOLOv8s [23]. Chen proposed an improved multitask deep convolutional neural network (DCNN) detection model based on YOLOv7 [24], which added two decoders to YOLOv7 for detecting tomato fruit clusters, fruit ripeness and cluster ripeness. Subsequently, the loss function was designed according to the characteristics of multi-tasking, and scale-sensitive joint intersection (SIoU) was used instead of complete joint intersection (CIoU) to improve the recognition accuracy of the model. An improved yellow peach detection model (EMA-YOLO) based on YOLOv8 is proposed, which reduces the leakage and false detection rate of target small yellow peaches by introducing an EMA attention mechanism module to encode global information [25]. |
||
Comments 6: Image enhancement should be applied to the figures in the paper to improve image clarity especially for Figures 3, 8, and 9. |
||
Response 6: Thank you for pointing this out. It has been modified as needed, and Figure 3,Figure 8 and Figure 9 have been HD processed. For more information, see lines 216, 378 and 405. |
||
Comments 7: The versions of YOLOv5 and YOLOv8 used for comparison in the paper are not specified, which may lead to ambiguity. It is suggested to provide the specific versions. |
||
Response 7: Thank you for pointing this out. Comparison experiments were designed to compare the improved model with the original versions of YOLOv5 and YOLOv8, and were supplemented with comparisons and descriptions with the base model YOLOv8s. The details of the modifications and additions are set out below. For more information, see lines 492 to 495. Under the same experimental environment, the improved YOLOv8s-GOLD-LSKA model in this study improved the F1 and mAP values by 5.42% and 6.63%, respectively, and the model size was reduced by 2.13 MB, as well as all other parameters, when compared with the base model YOLOv8s. Comments 8: Please clarify whether you plan to make your code publicly available. If it is released, it would be very helpful to interested readers. Response 8: Thank you very much for your suggestion. This project is currently in the development stage and has not yet been formally completed. In order to avoid copyright conflicts, after the publication of the paper, interested readers can contact the first author to explain the purpose of the use and obtain the code. |
Reviewer 3 Report
Comments and Suggestions for Authors
The paper proposes an enhanced sorghum spike detection model based on YOLOv8s for identifying sorghum spikes in natural field environments. However, there are still the following issues with the content of the article:
(1) Some images in the article are unclear and blurry, such as Figure 3 and Figure 9.
(2) In Section 2.1, there are two sections with identical text paragraphs.
(3) In Section 2.1, only the process of image acquisition is introduced, and a simple description of the dataset content is missing.
(4) In Section 2.1, Figure 2 lacks an introduction to data augmentation, which is instead introduced based on the image.
(5) In Section 2.2, different versions of the YOLOv8 model are mentioned, but no relevant literature is cited for reference.
(6) In Figure 6, the annotation of the symbol at the junction is missing.
(7) In Section 2.2.4, the explanation of the meanings of the individual symbols in the formula is incomplete.
(8) In Section 3.2, the description of the content of Table 2 is partially inaccurate.
(9) In Section 3.3, there is a lack of comparison with the basic YOLOv8s model, making it difficult to intuitively understand the superiority of the proposed method.
(10) In Section 4, the summary of the article is not comprehensive enough; the Focal-EIOU loss function used is not summarized, and the comparison with the YOLOv8s model results is not mentioned in the text.
Author Response
For research article
Response to Reviewer 1 Comments
|
||
1. Summary |
|
|
Thank you very much for taking the time to review this manuscript. Please find the detailed responses below and the corresponding revisions/corrections highlighted/in track changes in the re-submitted files. [This is only a recommended summary. Please feel free to adjust it. We do suggest maintaining a neutral tone and thanking the reviewers for their contribution although the comments may be negative or off-target. If you disagree with the reviewer's comments please include any concerns you may have in the letter to the Academic Editor.]
|
||
2. Questions for General Evaluation |
Reviewer’s Evaluation |
Response and Revisions |
Does the introduction provide sufficient background and include all relevant references? |
Can be improved |
Changes have been made in the manuscript |
Is the research design appropriate? |
Can be improved |
Changes have been made in the manuscript |
Are the methods adequately described? |
Can be improved |
Changes have been made in the manuscript |
Are the results clearly presented? |
Must be improved |
Changes have been made in the manuscript |
Are the conclusions supported by the results? |
Can be improved |
Changes have been made in the manuscript |
3. Point-by-point response to Comments and Suggestions for Authors |
||
Comments 1: Some images in the article are unclear and blurry, such as Figure 3 and Figure 9. |
||
Response 1: Thank you for pointing this out. It has been modified as needed, and Figure 3 and Figure 9 have been HD processed. For more information, see lines 216 and 407. |
||
Comments 2: In Section 2.1, there are two sections with identical text paragraphs. |
||
Response 2: Thank you for pointing this out. It has been modified as needed, and the text of the second paragraph has been modified to data set establishment and data amplification. The details of the modifications and additions are set out below. For more information, see lines 171 to 185. There are numerous complications in sorghum spike images taken in natural field environments such as heavy shading between sorghum spikes, dense distribution of sorghum spikes, and windy conditions that result in unclear captured images; therefore, the collected images need to be screened to remove those that are poorly captured and cannot be recognized and labeled. Considering the laboratory computer hardware and GPU performance, the image pixels are compressed to 1024 pixels × 768 pixels to speed up the model training. LabelImg software is used to manually label the data set according to the YOLO format, and the label file is saved as XML format. In order to alleviate the over-fitting problem caused by the small data set and improve the generalization ability of the model training results, Python is used to amplify the image data, and the sorghum data set is rotated, flipped, mirrored, and brightness adjusted, as shown in Figure 2. After data enhancement of the original sorghum spike data set, a total of 4500 images were obtained. The data set contains pictures of sorghum milk ripening period, wax ripening period and full ripening period taken from different angles, and the data set is divided into training set, verification set and test set according to the ratio of 8 : 1 : 1. |
||
Comments 3: In Section 2.1, only the process of image acquisition is introduced, and a simple description of the dataset content is missing. |
||
Response 3: Thank you for pointing this out. In the body of the second paragraph of Section 2.1, the description and introduction of the content of the dataset are added. The details of the modifications and additions are set out below. For more information, see lines 175 to 185. Considering the laboratory computer hardware and GPU performance, the image pixels are compressed to 1024 pixels × 768 pixels to speed up the model training. LabelImg software is used to manually label the data set according to the YOLO format, and the label file is saved as XML format. In order to alleviate the over-fitting problem caused by the small data set and improve the generalization ability of the model training results, Python is used to amplify the image data, and the sorghum data set is rotated, flipped, mirrored, and brightness adjusted, as shown in Figure 2. After data enhancement of the original sorghum spike data set, a total of 4500 images were obtained. The data set contains pictures of sorghum milk ripening period, wax ripening period and full ripening period taken from different angles, and the data set is divided into training set, verification set and test set according to the ratio of 8 : 1 : 1. |
||
Comments 4: In Section 2.1, Figure 2 lacks an introduction to data augmentation, which is instead introduced based on the image. |
||
Response 4: Thank you for pointing this out. In Section 2.1, the introduction of data augmentation is added to the text of the second paragraph. The details of the modifications and additions are set out below. For more information, see lines 178 to 182. In order to alleviate the over-fitting problem caused by the small data set and improve the generalization ability of the model training results, Python is used to amplify the image data, and the sorghum data set is rotated, flipped, mirrored, and brightness adjusted, as shown in Figure 2. After data enhancement of the original sorghum spike data set, a total of 4500 images were obtained. |
||
Comments 5: In Section 2.2, different versions of the YOLOv8 model are mentioned, but no relevant literature is cited for reference. |
||
Response 5: Thank you for pointing this out. Citation of references to different versions of YOLOv8 is added in section 2.2. For more information, see lines 198 and 210. |
||
Comments 6: In Figure 6, the annotation of the symbol at the junction is missing. |
||
Response 6: Thank you for pointing this out. It has been modified as needed, and annotations have been added under each picture. The details of the modifications and additions are set out below. For more information, see lines 279 and 281. (1)Low-GD (2)High-GD |
||
Comments 7: In Section 2.2.4, the explanation of the meanings of the individual symbols in the formula is incomplete. |
||
Response 7: Thank you for pointing this out. The meaning explanation of each symbol in Section 2.2.4 formula has been supplemented. The details of the modifications and additions are set out below. The details of the modifications and additions are set out below. For more information, see lines 381 to 384. Figure 8 illustrates the parameters involved in the calculation, wc and hc are the width and height of the minimum circumscribed rectangle of the anchor frame and the target frame. ρ is the Euclidean distance between b and bgt, and γ is the parameter to control the suppression degree of outliers. Comments 8: In Section 3.2, the description of the content of Table 2 is partially inaccurate. Response 8: Thank you for pointing this out. Inaccurate descriptions and data errors in Table 2 have been corrected and amended in Section 3.2. For more information, see lines 447, 450 and 468. Comments 9: In Section 3.3, there is a lack of comparison with the basic YOLOv8s model, making it difficult to intuitively understand the superiority of the proposed method. Response 9: Thank you for pointing this out. The comparison with the basic model YOLOv8s has been added in Section 3.3. The details of the modifications and additions are set out below. For more information, see lines 492 to 495. Under the same experimental environment, the improved YOLOv8s-GOLD-LSKA model in this study improved the F1 and mAP values by 5.42% and 6.63%, respectively, and the model size was reduced by 2.13 MB, as well as all other parameters, when compared with the base model YOLOv8s. Comments 10: In Section 4, the summary of the article is not comprehensive enough; the Focal-ELOU loss function used is not summarized, and the comparison with the YOLOv8s model results is not mentioned in the text. Response 10: Thank you for pointing this out. In Section 4, the summary of the paper is refined, adding a description of the Focal-EIOU loss function and a comparison of the improved model with the YOLOv8s model. The details of the modifications and additions are set out below. For more information, see lines 515 to 523. The model incorporates the GOLD feature pyramid module and the LSKA attention mechanism and Focal-EIOU loss function to address the challenge of detecting difficult targets in such contexts. The improved YOLOv8s-GOLD-LSKA model achieves a Precision value of 90.72%, a Recall value of 76.81%, an F1 score of 81.19%, a mean Average Precision (mAP) of 85.86%, a model size of 7.48 MB, and an average detection time of 0.0168 seconds per image. Compared with the basic model YOLOv8s, the F1 value and mAP value were increased by 5.42 % and 6.63 % respectively, the model size was reduced by 2.13 MB, and other parameters were also improved. |
Round 2
Reviewer 2 Report
Comments and Suggestions for Authors
The author has made some revisions in response to the review comments, but the following major issues remain. I believe this article should be re-evaluated:
1. The revisions addressing Comment 3 lack supporting experimental data and therefore fail to meet the requirements.
2. In response to Comment 6, Figure 8 contains Chinese characters, and Figure 9 remains unclear.
3. Regarding the revisions for Comment 7, the specific version number of YOLOv5 used has still not been clearly indicated.
4. In addressing Comment 8, the source code has not been made publicly available, which will hinder the re-producibility of the experiments.
Author Response
For research article
Response to Reviewer 3 Comments
|
||
1. Summary |
|
|
Thank you very much for taking the time to review this manuscript. Please find the detailed responses below and the corresponding revisions/corrections highlighted/in track changes in the re-submitted files. [This is only a recommended summary. Please feel free to adjust it. We do suggest maintaining a neutral tone and thanking the reviewers for their contribution although the comments may be negative or off-target. If you disagree with the reviewer's comments please include any concerns you may have in the letter to the Academic Editor.]
|
||
2. Questions for General Evaluation |
Reviewer’s Evaluation |
Response and Revisions |
Does the introduction provide sufficient background and include all relevant references? |
Can be improved |
Changes have been made in the manuscript |
Is the research design appropriate? |
Yes |
|
Are the methods adequately described? |
Can be improved |
Changes have been made in the manuscript |
Are the results clearly presented? |
Must be improved |
Changes have been made in the manuscript |
Are the conclusions supported by the results? |
Can be improved |
Changes have been made in the manuscript |
3. Point-by-point response to Comments and Suggestions for Authors |
||
Comments 1: The revisions addressing Comment 3 lack supporting experimental data and therefore fail to meet the requirements. |
||
Response 1: Thank you for pointing this out. Regarding the modification in your comment 3, there is a lack of data support for the detection effect in different scenarios, as indicators such as shooting angle were not uniformly divided during the process of dividing the dataset. However, based on the description of the detection results of sorghum ears under different shooting angles in section 3.2, additional explanations are provided. The details of the modifications and additions are set out below. For more information, see lines 483 to 486. The accuracy of detection based on pictures is not high due to severe occlusion between sorghum spikes in images taken from the front. Therefore, detection can be performed based on images taken from directly above, avoiding the occlusion problem and providing better detection results. |
||
Comments 2: In response to Comment 6, Figure 8 contains Chinese characters, and Figure 9 remains unclear. |
||
Response 2: Thank you for pointing this out. Figure 8 has been corrected and Figure 9 has been high-resolution. For more information, see lines 378 and 405. |
||
Comments 3: Regarding the revisions for Comment 7, the specific version number of YOLOv5 used has still not been clearly indicated. |
||
Response 3: Thank you for pointing this out. Additions and corrections have been made to the specific version of YOLOv5 in the article. The details of the modifications and additions are set out below. For more information, see lines 495 to 507. With other classical models YOLOv5s, SSD and YOLOv8 tested on a self-constructed sorghum spike dataset. |
||
Comments 4: In addressing Comment 8, the source code has not been made publicly available, which will hinder the re-producibility of the experiments. Response 4: I have seriously considered your suggestion. I am very sorry that this project is currently in the development stage and has not yet been officially completed, and the author has not been allowed to disclose the data. When the patent application is successful, the code will be uploaded to GitHub. hope to gain your understanding. |
Reviewer 3 Report
Comments and Suggestions for Authors
accept
Author Response
Thank you very much for taking the time to review this manuscript.
Round 3
Reviewer 2 Report
Comments and Suggestions for Authors
All comments were addressed